# 3D ophthalmic ultrasonography at the slit lamp using existing ultrasound systems

Jack O. Thomas[1,2☯], Josiah K. To[1☯], Parsa Riazi Esfahani[1,2], Frithjof Kruggel[3], William C. Tang[3], Andrew W. Browne[1,3,4,5]*

1 Department of Ophthalmology, Gavin Herbert Eye Institute, University of California Irvine, Irvine, California, United States of America, 2 School of Medicine, California University of Science and Medicine, Colton, California, United States of America, 3 Department of Biomedical Engineering, Henry Samueli School of Engineering, University of California Irvine, Irvine, California, United States of America, 4 Gavin Herbert Eye Institute, Center for Translational Vision Research, University of California Irvine, Irvine, California, United States of America, 5 Institute for Clinical and Translational Science, School of Medicine, University of California Irvine, Irvine, California, United States of America

☯ These authors contributed equally to this work.
* abrowne1@hs.uci.edu

## Abstract

### Purpose

This study aims to explore the feasibility and performance of three-dimensional ultrasound (3DUS) imaging in ophthalmology using commercially available ultrasound probes adapted to a slit lamp.

### Significance

Despite ultrasound's long-standing application in eye care for visualizing ocular components, the evolution of 3DUS technology has remained inactive, with limited development and commercial availability. This study introduces a novel method that could potentially enhance ophthalmic diagnostics and treatment planning by providing comprehensive 3D views of ocular structures using existing ultrasound probes adapted to the conventional slit lamp.

### Methods

A custom system was designed for mounting a linear actuator to a slit lamp and enabling the horizontal actuation of any ultrasound probe. Ophthalmic and non-ophthalmic ultrasound probes were tested for their ability to reconstruct 3DUS images of the posterior pole. The study involved designing and evaluating three US phantoms ex vivo and performing in vivo imaging on human subjects to assess the system's applicability.

### Results

The system successfully acquired 3D volume scans of phantoms and live human eyes, demonstrating the system's potential for detailed ocular imaging. The adaptability of the device allowed for compatibility with various ultrasound probes. In vivo imaging revealed the

**Data Availability Statement:** All relevant data are within the manuscript.

**Funding:** The author Andrew W. Browne (AB) received funding from the following sources to help

support this study: NIH/NEI 1K08EY034912 - 01, unrestricted Research to Prevent Blindness award to the University of California Irvine Dept. of Ophthalmology, International Retinal Research Foundation, and BrightFocus Foundation. The funders had no role in study design, data collection and analysis, decision to publish, or preparation of the manuscript.

**Competing interests:** The authors have declared that no competing interests exist.

system's capability to produce high-resolution 3D reconstructions of ocular structures, including eyes with pathological conditions.

## Conclusions

The introduction of a slit lamp-mounted 3DUS system represents a significant advancement in ophthalmic ultrasonography, offering a practical and accessible solution for comprehensive 3D ocular assessments. The device's modularity and compatibility with existing ultrasound probes make it a versatile tool for a wide range of ophthalmic applications. Further research and clinical trials are needed to optimize the system's performance and validate its clinical utility.

## Introduction

Ultrasound (US) technology has consistently been present in medical diagnostics, offering non-invasive, real-time imaging capabilities across various clinical settings. In ophthalmology, US visualizes ocular components, particularly when optical clarity is compromised. Despite its long-standing use, the evolution of US technology in eye care has been limited over the past few decades. The introduction of 3D ultrasound (3DUS) in the 1990s represented a notable development, providing more comprehensive views of ocular structures. This technology has been applied in various medical fields, as demonstrated by numerous studies.

In ophthalmology, 3DUS was introduced using a custom rotational actuator to rotate a US probe on the eye and generate a 3D volume of 2-dimensional (2D) B-scan images of the posterior pole -[1]. As of the publication date of the present paper, this original publication was found to be the only reported study to employ rotational actuation of the US probe on the eye. Linear actuation of US probes was used in almost all subsequent studies. Cusumano et al. in 1998 used a linear actuator to scan US probes across the eye to study various conditions [2]. This preliminary work was followed by other studies that examined either anterior segment, posterior segment, or optic nerve disease.

The application of 3DUS in the anterior segment was achieved using a US biomicroscopy (UBM) probe to characterize the ciliary body anatomy, the posterior iris surface, and changes in the ciliary body with accommodation [3–5]. Most recently, Helms et al. and Minhaz et al. used linear scan actuation with a 50 MHz UBM probe for characterizing the anterior segment in ex vivo human eyes, living rabbits, and in eyes before and after transscleral cyclophotocoagulation [6, 7]. These studies provide detailed images of the front part of the eye, potentially aiding in diagnosing and managing anterior segment pathologies.

In retinal imaging, the only commercially available 3DUS system (3D i-scan® from Ophthalmic Technologies Inc, Toronto, Ontario) was used to study radioactive plaque placement over choroidal melanomas [8] and choroidal melanoma volume [9]. 3D i-scan® was also used to evaluate the posterior pole of a uveitis patient with no optical view to the retina [10]. A custom 20 MHz 5-ring annular probe introduced the ability to synthetically focus from anterior to posterior segments to render 3D models from linear scans using one probe [11]. 3DUS has been used to evaluate the retroorbital optic nerve diameter using the i-scan in pilot studies and high-altitude cerebral edema in hikers climbing Mount Everest [12, 13]. While these other studies employed ophthalmic US probes, non-ophthalmic US probes have been used to evaluate eyelid anatomy, cornea, iris, the entire eye, and temporal artery blood flow [14]. In a recent study, the efficacy of 2D and 3DUS in objectively estimating retinoblastoma primary tumor

volume was evaluated, showing that 3DUS, especially with volumetric probes, provides a convenient and accurate method of measurement amid the irregular morphology of the retinoblastoma primary tumor [15]. These studies demonstrate the use of 3DUS in providing volumetric and 3D structural information about intraocular and extraocular contents.

While 3D ophthalmic US was introduced nearly 30 years ago, only one commercial device was developed and did not persist in the market. Currently, 3DUS technology is not accessible in regular clinical practice. There is no known instance of a US probe being mounted on an apparatus to replicate the optical imaging experience with a chinrest-style setup akin to other ophthalmic imaging methods such as keratometry, optical coherence tomography (OCT), or fundus photography. Additionally, while there have been recent investigations into the application of non-ophthalmic US probes in detecting ocular pathology, none have utilized 3DUS [16].

Given the absence of 3DUS technology in ophthalmology and the failure of prior commercial instruments, our research aimed to explore the feasibility and performance of 3DUS at the slit lamp using commercially available US probes. To achieve this, we developed a custom horizontal actuation system suitable for any US probe, mounted it to a slit lamp, and used three different ophthalmic and non-ophthalmic US probes to reconstruct 3DUS images of the posterior pole.

## Discussion/Conclusions

Our work represents a novel approach to extend the capability of existing 2D US hardware to achieve 3DUS imaging in a more accessible and clinically relevant format, potentially expanding its utility in ophthalmic diagnostics and treatment planning.

## Methods

### Slit lamp-mounted US linear actuator

A custom stand for mounting a linear actuator to a BQ900 slit lamp (Haag-Streit AG, Köniz, Switzerland) was designed using Autodesk Fusion 360® (Autodesk, Inc., San Rafael, CA) based on precise measurements of the slit lamp obtained with a digital caliper (Fig 1). The design was 3D printed on a Prusa i3 MK3S (Prusa Research, Prague, Czech Republic) using polyethylene terephthalate glycol (PETG) filament, with print settings adjusted to a 0.2 mm layer height, 20% infill density, and 60 mm/s print speed, followed by necessary post-processing for a smooth finish and fitment. The linear actuator was secured to the 3D printed stand so that it would move in a path parallel to the person's face positioned in the slit lamp's chin rest. A custom-designed 3D-printed PETG bracket to hold different US probes was produced and secured to the linear actuator's sliding element. The final platform was then securely attached to the slit lamp, ensuring compatibility and functionality. The linear actuator was controlled using an Arduino® microprocessor (Arduino LLC, Somerville, MA) and stepper motor drivers, facilitating precise movements required for diagnostic procedures.

### Phantoms

Three US phantoms were developed and analyzed to evaluate image volume capture ex vivo. The first phantom created was a 3D-printed polylactic acid (PLA) grid on a Prusa i3 MK3S+® (Fig 2). The second phantom featured an indented spherical grid, which was 3D printed using elastic polymer resin (Flexible 80A V1) on a Form 3B® printer (FORMS LABS, City, State) (Fig 2). The third phantom was created to match the structure of the resin phantom by pouring wax into a negative 3D-printed mold. Each phantom was submerged in Knox Gelatin®

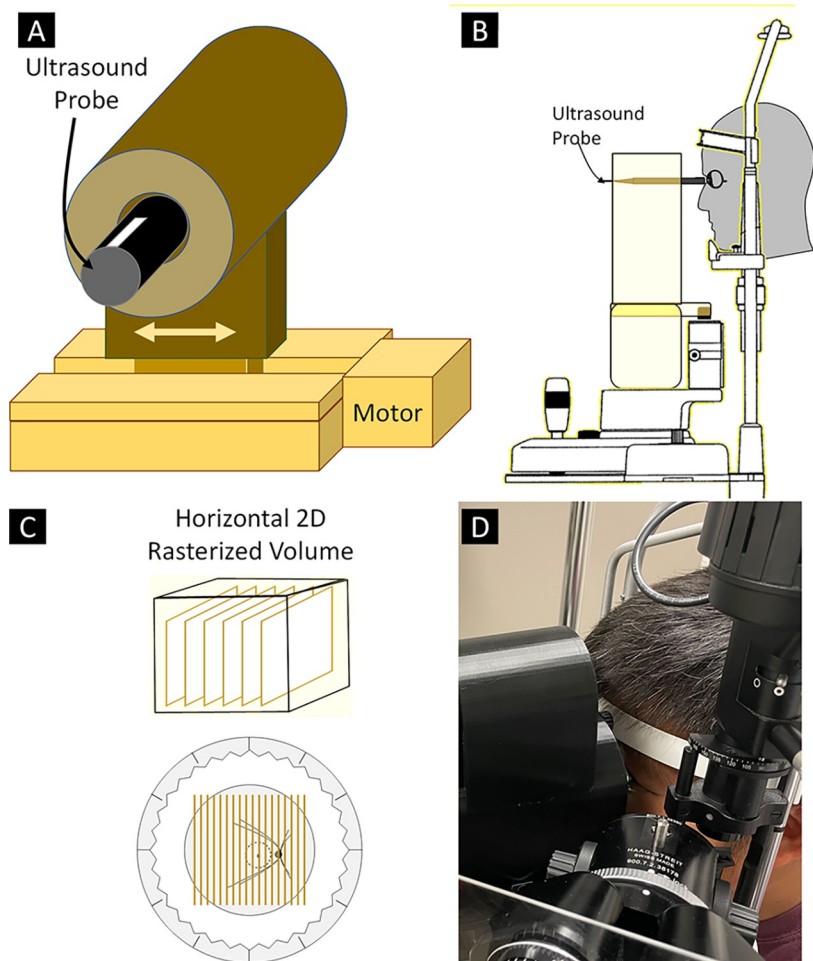

**Fig 1. Slit lamp-mounted linear actuator. (A)** Diagram of linear actuator with US probe holder. **(B)** Diagram of slit lamp-mounted linear actuator. **(C)** Horizontally arrayed 2D B-scan images produced by linear actuation parallel to the facial plane. **(D)** Photograph of linear actuator holding probe on subject's eye while positioned in a slit lamp.

mixed with a gelatin-to-water ratio of 5:1 by weight to simulate the acoustic properties of vitreous humor and refrigerated at 5˚C for at least 10 hours [17].

## Image acquisition

Three US probes were evaluated in this study: Accutome B-Scan Pro® 12 MHz probe (A12) (Keeler, Malvern, PA, USA), Quantel ABSolu® annular 20 MHz probe (AQ20) (LUMIBIRD, Cesson-Sévigné, France), and the Butterfly iQ® (BiQ) (Butterfly Network Inc, Guilford, CT, USA). The A12 and AQ20 probes acquired all data using default settings, while the BiQ was used exclusively with the ophthalmic preset function. Technical specifications for each of these US probes are summarized in the Supporting Information section (S1 Table).

Each US probe was utilized using the stock software provided by each probe manufacturer. Videos of each US probe were recorded in the B-scan setting as each probe was moved in a linear path across a phantom or subject's eye. Phantom imaging involved linear scans of the three US probes at a fixed speed (1 mm/s) across the phantoms. The wax phantom was also scanned at 2, 4, and 9 mm/s with the A12 probe. Clear gelatin phantoms, created in the same fashion as described above, were imaged using the device to assess noise levels. Non-border regions were

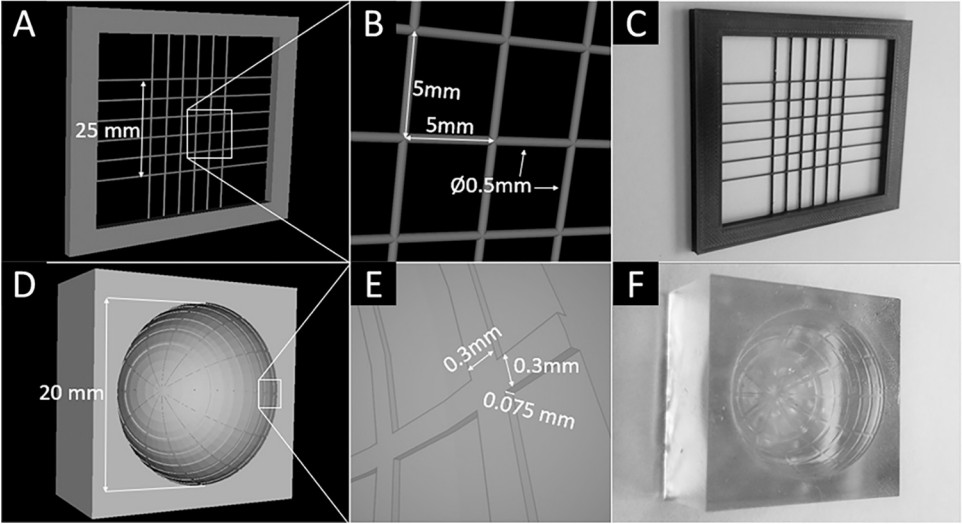

**Fig 2.** 3D models of planar (A, B) and spherical (D, E) grid phantoms. Photographs of planar (C) and spherical (F) grid phantoms.

then sampled and analyzed using ImageJ to determine Mean Signal (μ) and Standard Deviation (σ, Noise) (S1 Fig).

Institutional Review Board approval was obtained from the University of California, Irvine, (IRB #20195254) for clinical data collection. The study adhered to the Declaration of Helsinki, and written informed consent was obtained from 44 participants in a HIPAA-compliant manner. The recruitment period for this study was from February 13th, 2023 to June 1st, 2023. Patients received Tetracaine Hydrochloride Ophthalmic Solution, USP 0.5% (Alcon, Geneva, Switzerland) eye drops, were positioned in a slit lamp, and GenTeal Gel (Alcon, Geneva, Switzerland) was applied to the US probe as a lubricating agent. The probe was brought into contact with patients' corneas, and horizontal scans were obtained.

## Image processing

Frames from the US video files were extracted as 2D images, cropped, and then converted into a sequence of PNG files for offline processing [18]. Sequential pairs of images were aligned using linear registration to correct for a possible drift in the US probe position. Image annotations were detected as high-intensity pixels that were constant throughout the time series and, thus, removed. Finally, the set of 2D images was linearly interpolated [19] in 3D space using the known angular motion of the probe and stored as a Nifti file.

## Results

### Phantom imaging

We acquired 3D volume scans of three phantoms to evaluate 3D imaging using US probes from different manufacturers. The wax phantom was imaged by translating the A12 parallel to the phantom at different linear actuation speeds (Fig 3). When the probe was moved rapidly across the phantom at 9 mm/sec, the reconstructed model had stepwise artifacts that reduced model smoothness. When the probe moved 2mm/sec or slower, the 3D model appeared smoother.

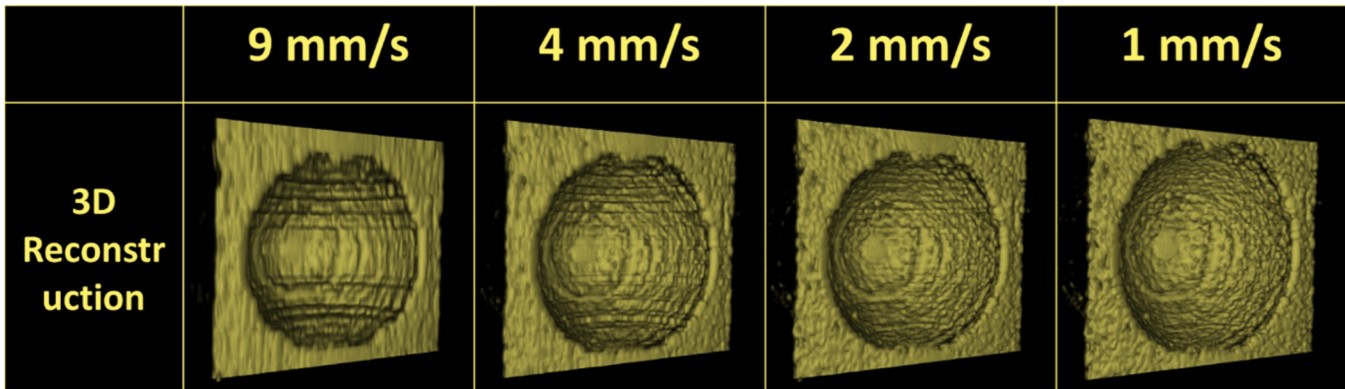

**Fig 3. 3D models produced from actuating the A12 at different speeds parallel to the wax phantom.**

We subsequently evaluated model reconstruction for different probes using the PLA grid phantom (Fig 4A) and the resin phantom with smooth continuous and microscopic features (Fig 4B).

Original B-scan images of the PLA grid phantom using each US probe produced an artifact where features central to the US probe appeared flat and perpendicular to the US propagation waves. In contrast, features at the periphery of the US probe field of view appeared angled. This phenomenon was highlighted with lines drawn below each feature on the original B-scan images (Fig 4A). By overlying these lines with each other (bottom row of Fig 4A), we observed that the A12 probe produced a sizeable off-axis angulation artifact. By connecting a line from the centroids of each feature, we observed a curvature artifact for the planar grid. The curvature artifact was lowest for the BiQ. The length of the line connecting feature centroids also varied. The BiQ produced the shortest line, while the A12 produced the longest line.

Reconstructions of the resin phantom scanned with the AQ20, A12, and BiQ are shown in Fig 4B. All three US probes identified raised concentric circles in the resin phantom on the original B-scan and the resultant 3D reconstructions. The AQ20 resolved the resin phantom's concentric circles with thin lines in a nearly 360-degree fashion. While the A12 showed the concentric rings, the rings were not as easily distinguishable. The BiQ likewise resolved the concentric rings. However, with the BiQ, the concentric rings in the 3D reconstruction appeared larger than the features of the actual phantom. Furthermore, while the resin phantom contained 6 concentric rings, the AQ20 and A12 only resolved 5 of the rings, and the BiQ was able to resolve only 4 of the rings.

## In vivo imaging

Next, to evaluate the 3DUS device on live human subjects, we acquired scans of a subject's healthy eye with the AQ20, A12, and BiQ (Fig 5).

The larger dimensions of the BiQ probe significantly limited the distance of the horizontal scan because as the probe transversed the globe, it would contact the orbital bones (Fig 5). Therefore, the AQ20 and A12 probes could travel longer distances while scanning horizontally than the BiQ.

The 3D reconstruction from the AQ20 (Fig 5) appeared qualitatively to have an increased resolution compared to the A12 and BiQ. While the scan with the A12 resulted in a 3D construction similar in morphology and curvature to the AQ20, the structures appeared less sharp and resolved than the AQ20. Both AQ20 and A12 showed features resembling temporal vascular arcades emanating from an optic nerve sheath. The 3D reconstruction using the BiQ did

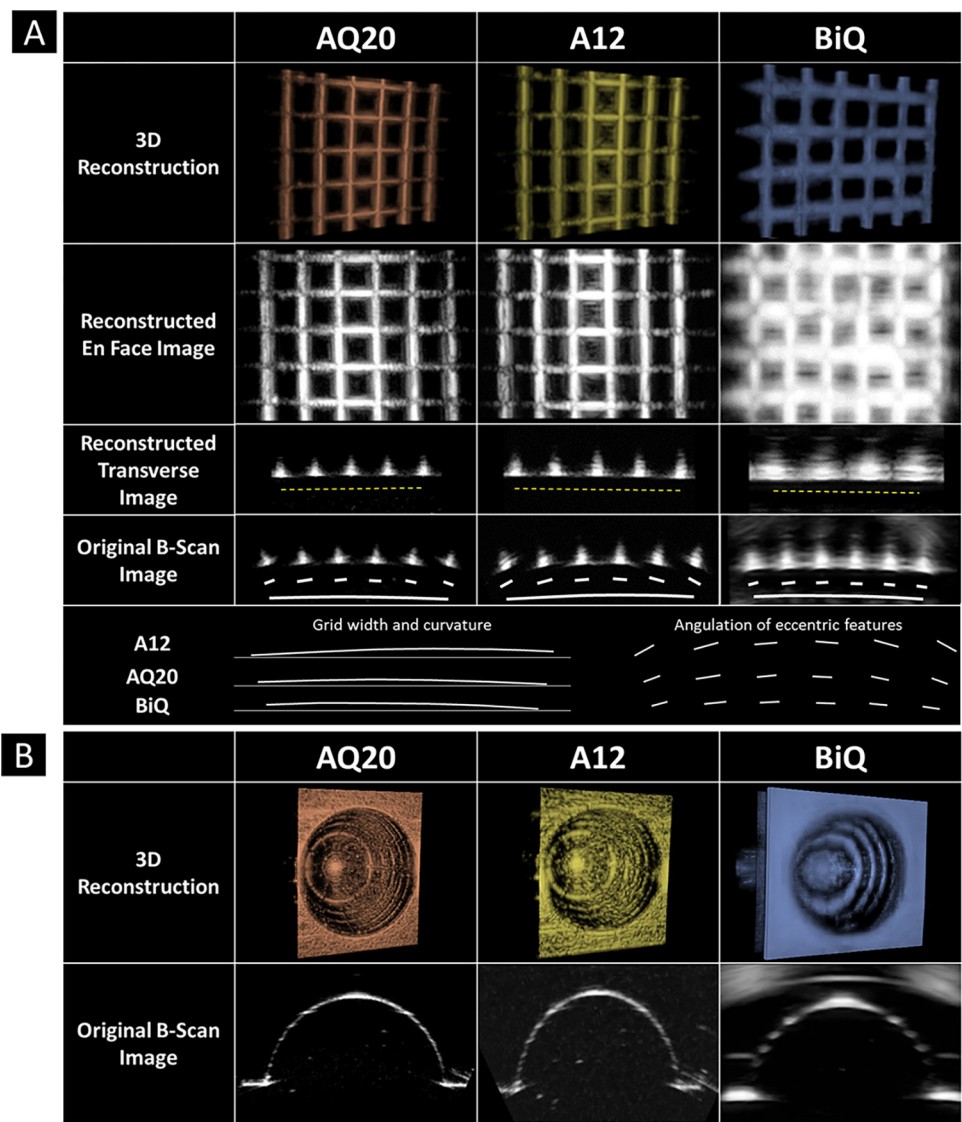

**Fig 4. Resultant 3D reconstructions of planar grid and resin phantoms. (A)** 3D reconstructions, en face, transverse, and original B-scan images of the planar grid phantom imaged with the AQ20, A12, and BiQ. Dashed yellow lines were drawn below reconstructed transverse images, showing the flatness of the reconstructed view. Curved lines were drawn below the original B-scan images to highlight the artifactually appearing curvature of the flat grid. Line segments were drawn parallel to each grid feature to show the artifactual angulation of features in the eccentric parts of the original B-scan image. **(B)** 3D reconstructions and original B-scan images of the resin phantom with the AQ20, A12, and BiQ, demonstrating the concentric and radial grid features in each reconstruction.

not reveal significant information about the morphology of the posterior structures of the eye, and its resolution appeared to be limited compared to the AQ20 and A12. Original B-scan images from BiQ showed anterior and posterior segment structures, while AQ20 and A12 showed only posterior pole structures.

We subsequently imaged two subjects with clinically and structurally obvious abnormalities. One patient who came for a second opinion for pathological myopia had a history of vitrectomy and epiretinal membrane peeling. OCT showed posterior staphyloma with geographic atrophy and a chronic eccentric macular hole (Fig 6A). The 3DUS reconstruction of the subject with staphyloma (Fig 6C) revealed a defined staphyloma that was less clearly

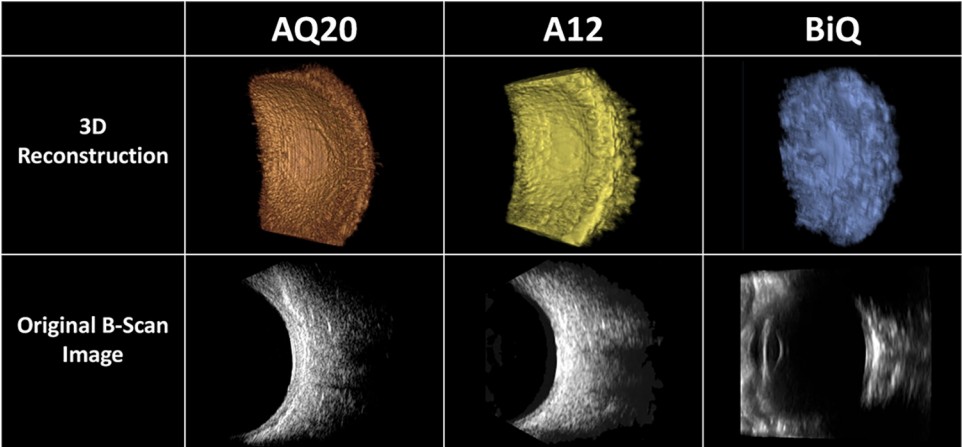

**Fig 5. 3D reconstructions and B-scan images of a healthy human eye using AQ20, A12, and BiQ probes.**

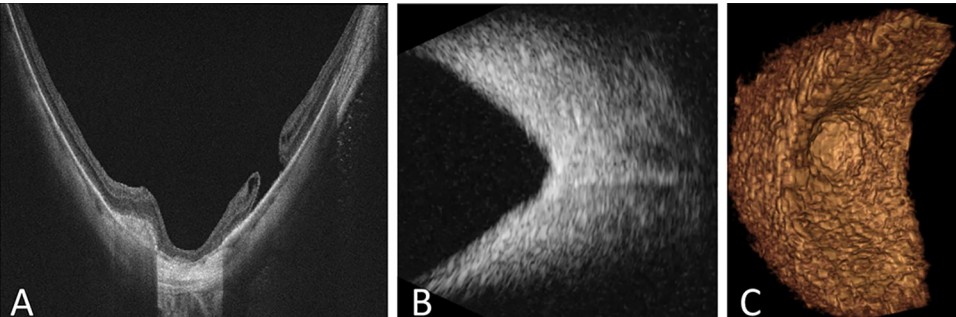

**Fig 6.** Patient with pathological myopia with staphyloma demonstrated on OCT (A), B-scan US (B), and 3DUS (C).

delineated on a single 2D B-scan image (Fig 6B). A second patient came for a second opinion after silicone oil removal following a history of endophthalmitis complicated by tractional retinal detachment and was treated with scleral buckle placement and silicone oil. We obtained ultra-widefield fundus photography prior to oil removal that showed scleral buckle indentation with laser scars and subretinal perfluoro-n-octane (PFO) bubbles (Fig 7A). 3DUS reconstruction showed copious residual intraocular silicone oil droplets. The spatial relationships

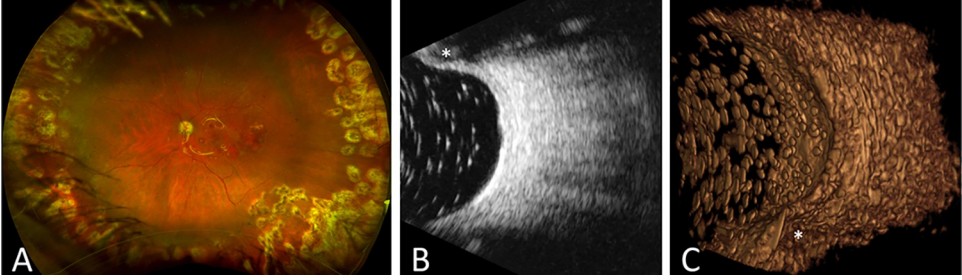

**Fig 7.** Patient with history of tractional retinal detachment repaired with scleral buckle and silicone oil as seen in widefield fundus photograph (A). After silicone oil removal, residual oil droplets are seen on B-scan ultrasound (B) and 3DUS (C). Scleral buckle is indicated by an asterisk in (B) and (C).

between the opacities that were not obvious from individual B-scan images (Fig 7B) were more clearly visualized by the 3DUS reconstruction (Fig 7C).

## Discussion

The published history of 3D ophthalmic US began in the 1990s, marking a significant advancement in eye care diagnostics by offering comprehensive views of ocular structures by aggregating 2D B-scan images into 3D volumes [1]. Initial applications utilized custom and linear actuation techniques for probe movement, enabling detailed studies of both anterior and posterior eye segments, as well as the optic nerve, across various conditions and treatments [2–7]. Despite its potential, the development and commercial availability of 3D ophthalmic US devices have been limited, with only one commercial device briefly entering the market [8, 9, 12, 13], leaving this technology largely inaccessible in routine clinical practice.

In this study, we introduced a novel approach for adapting existing US probes to a slit lamp, enabling the acquisition of noninvasive, rapid 3DUS reconstructions of ophthalmic structures. Several features of this device and approach represented an advancement in the field of ocular 3DUS.

First, our device demonstrated the practicality of a 3D ocular US mounted onto a slit lamp. The device being compatible with a slit lamp could allow for a seamless introduction into ophthalmology practices, as slit lamps are ubiquitous in this setting. Notably, the i-scan®, the only commercially available 3D ocular US, lacked this feature. Additionally, our device capitalized on the stability provided by a chinrest setup, a standard feature in various ophthalmologic imaging tools like OCT, fundus photography, and the slit lamp itself. The 3D ocular US device required approximately 10 seconds per scan. Each imaging session required less than 5 minutes, which would make it a feasible enhancement to the standard ophthalmologic workup if further studies show clinical utility. Ultimately, the standardized acquisition protocols that mirror optical imaging would extend the ability of a technician to acquire US images that would otherwise require a trained ophthalmologist or ophthalmic ultrasonographer.

Second, our 3D ocular US device was designed to be modular, allowing it to accept multiple ophthalmic and non-ophthalmic US probes. In this study, the versatility of the device was demonstrated by using two ophthalmic US systems and a non-specific US probe. This capability significantly enhanced the flexibility of the device, enabling it to accommodate probes with various imaging characteristics, thereby substantially expanding its potential applications and adaptability. Anterior segment imaging and ophthalmic imaging in the emergency room introduce two additional contexts where slit lamp-based imaging and the ability to accommodate different probes may benefit from our approach [6, 7, 20].

In this study, we demonstrated the capability of the 3D ocular US device to create 3D models of three distinct phantoms. The high density of printable polymers inherently limits manufacturing precision of US phantoms with small features using commercial 3D printers. All materials, from silicone and wax to extruded PLA and printed resins have ultrasonic properties that mimic bone more than soft tissues [21]. Commercial phantoms are cost-prohibitive, and the smallest features in commercial phantoms are produced by polymer strings passing through a hydrogel matrix and producing results similar to those of high-density printable materials.

Scanning the wax phantom at varying speeds with the A12 revealed a significant tradeoff between the scan speed and the resulting quality of the 3D reconstruction (Fig 3). This was particularly evident with fast scan speeds, where pronounced stepwise artifacts were noticeable in the 3D model. These artifacts progressively diminished at speeds of 4 mm/s, 2 mm/s, and 1 mm/s. This phenomenon was compounded by the fixed framerate of the US probe, allowing a

set number of images to be captured per second. Consequently, as scanning speed increased, fewer images contributed to the 3D reconstruction, leading to reduced resolution and the apparent emergence of stepwise artifacts. While US images could successfully be extracted from A12 videos at 11 frames per second without duplication of data, images could only be extracted from AQ20 at 8 frames per second. Lower framerate imaging systems require longer scan durations to achieve the same number of image frames per scan length than higher frame rate ones. Regarding the clinical relevance of scan speed, the dimensions of pathological features could influence the scan speed parameters; more prominent features like retinal detachments may not require ultrahigh resolution scans to reliably detect in a series of images, while smaller features like retinal tears may demand lower scanning speed to reliably visualize in a 3D model. In addition to image quality tradeoffs, patient comfort should be considered. Prolonged scan durations also increase the likelihood of small patient movements introducing movement artifacts. Further investigation is needed to optimize scan parameters for each probe and imaging context.

Emphasis was placed on qualitative analysis in this pilot investigation primarily to demonstrate the potential utility of our 3D ocular US device. Larger-scale clinical studies that acquire image and model interpretation data will help demonstrate the clinical performance of our 3D ocular US device to produce 3D models of ophthalmic anatomy using different US probes. The AQ20, A12, and BiQ successfully imaged features of the PLA grid (Fig 4A). Qualitatively, the AQ20 and A12 3D reconstructions of the PLA grid were similar. However, the BiQ reconstruction displayed horizontal and vertical features of the grid that appeared considerably thicker than the actual phantom's features. However, this qualitative analysis is constrained by the absence of quantitative measurements on the 3D reconstructions. Imaging the resin phantom revealed that the AQ20, A12, and BiQ successfully captured the gross anatomical contour and 0.3 mm wide concentric rings transverse to the imaging plane. In contrast, radial features spanning the resin mold's circumference were less easily visualized from 3D renderings (Fig 4B). Models scanned with the higher frequency AQ20 probe exhibited better resolution compared to the A12, successfully identifying five out of six concentric rings in the resin phantom, whereas the A12 rendered the rings less clearly. The BiQ, while capturing four of the six rings, depicted them as thicker than they are. Radial lines were more discernible in the AQ20 scans compared to the A12 and barely visible in BiQ-produced models. The differences between BiQ and conventional ophthalmic US probes may arise from differences in US technology (CMUT vs piezoelectric transducers) and ultrasonic frequency.

Furthermore, this study demonstrated the application of the 3D ocular US device in the clinical setting, generating 3D models of both healthy human eyes and eyes with pathology. 3D renderings of a healthy human eye were generated using the AQ20, A12, and BiQ (Fig 5) as a reference comparator between US systems and diseased human eyes. Ocular disease states imaged in this study included myopic staphyloma and silicone oil droplets (Figs 6 and 7). The 3D rendering of the staphyloma highlighted the capability of 3D imaging to visualize the abnormal contours of myopic globes (Fig 6C). Unlike 2D B-scan images that rely on mental interpolation, the 3D image enabled direct visualization of the staphyloma's unique structure. The 3D reconstruction of silicone oil inclusions demonstrated the advantage of 3D imaging in visualizing the vitreous space more clearly (Fig 7B, 7C). While 2D B-scans showed vitreous opacities, 3D imaging revealed their spatial relationships more distinctly. We noted artifacts in 3D images where silicone oil droplets, which were nearly perfect spheroids, appeared as oblate spheroids. These artifacts varied in size, appearing larger and elongated in the anterior vitreous and smaller towards the posterior. This observation, along with angled reflectance in flat grid phantom images, indicated the need for improved image processing models to correct geometrical aberrations in US imaging.

This clinical data provided insight into how 3DUS can enhance visualization of the ocular anatomy. This could have utility in multiple clinical situations, including characterizing complex intraocular anatomy in tractional retinal detachments, objectively quantifying the degree of optic nerve cupping in patients with glaucoma, and in patients with no optical view to the posterior pole. Having 3DUS in these clinical scenarios may support surgical procedures aiming to improve vision. 3DUS quantifying optic nerve volume in diseases with elevated intracranial pressure or evaluating extraocular muscle anatomy in diseases like thyroid eye disease could also offer diagnostic and therapeutic aid without the need for magnetic resonance imaging or computerized tomography.

While these preliminary findings are promising, the 3D ocular US device has limitations. One challenge identified was the difficulty some subjects experienced in remaining still during the scan, a critical factor in preventing significant motion artifacts. Despite topical anesthesia and lubrication, some subjects reported discomfort with the contact of the US probe on the globe, leading to an inferior patient experience compared to non-contact imaging techniques like OCT. Subjects typically required several scans to acquire usable data free from significant motion artifacts. To minimize motion artifacts, future iterations of the device should investigate increasing scan speed, implementing real-time motion tracking and correction, enhancing patient stabilization with improved support systems, exploring redundant scanning paths, cross-referencing with other imaging modalities, and advanced post-processing techniques. To improve patient comfort, future studies with manufacturers may further optimize the probe-to-tissue interface with novel immersion-based methods and enhance corneal anesthesia and lubrication.

Furthermore, specific US probes demonstrated limited effectiveness with the device. For instance, the BiQ faced severe limitations in acquiring horizontal scans due to the large dimensions of the probe, leading to contact with the orbital bones. Limitations in patient comfort and obstructive surrounding anatomy support a shift from horizontal to rotational probe actuation. However, volumetric reconstruction from radially aligned images would be more challenging than volume rendering from horizontally aligned images.

Noise analysis of the US probes showed varying signal variability in gelatin, a tissue-like medium (S1 Fig). Although the primary aim of this study was not to quantify each probe's intrinsic signal-to-noise ratio, future research may explore its impact on 3D reconstruction of real-world pathological features.

Further studies are needed to develop quantitative methods to analyze 3D models of customized US phantoms to refine accuracy in 3D model reconstruction. While the study highlights the clinical potential of the 3D ocular US device by presenting normal and pathological human eyes in vivo, drawing definitive conclusions regarding its clinical utility over traditional 2D US demands clinical investigation in larger cohorts of patients across a wider variety of ocular conditions. Clinically, the 3D ocular US device would benefit from a graphical user interface that would allow for more seamless acquisition and recording of data. Further studies with larger 3D imaging datasets could employ machine learning to automatically segment ocular structures of interest, detect ocular pathologies, and potentially adjust for motion and other artifacts.

## Conclusion

Our study presented a novel approach to 3DUS imaging in ophthalmology. We demonstrated the adaptability of existing US probes to acquire US volumetric images at the slit lamp. The integration of our apparatus with slit lamps offered a practical and accessible solution for obtaining 3D images in a clinical setting and could standardize imaging protocols performed

by technicians. The potential applications span from posterior segment imaging to anterior segment and orbital assessments, showcasing the versatility and clinical relevance of our 3D ocular US apparatus. Further research and clinical validation will be crucial for optimizing volume rendering and investigating the device's utility in ophthalmic diagnostics and patient care.

## Supporting information

**S1 Table. Technical specifications for the Accutome B-Scan Pro, ABSolu 20 MHz annular ring probe, and Butterfly iQ (ophthalmic preset).**
(TIF)

**S1 Fig. Noise analysis of clear gelatin phantom imaging with the Accutome B-Scan Pro, ABSolu 20 MHz annular ring probe, and Butterfly iQ (ophthalmic preset).** Mean Signal (μ) is labelled in blue with Standard Deviation (σ, Noise) as error bars.
(TIFF)

## Author Contributions

**Conceptualization:** Jack O. Thomas, Josiah K. To, Parsa Riazi Esfahani, Frithjof Kruggel, William C. Tang, Andrew W. Browne.

**Data curation:** Jack O. Thomas, Josiah K. To, Parsa Riazi Esfahani, Frithjof Kruggel, William C. Tang, Andrew W. Browne.

**Formal analysis:** Jack O. Thomas, Josiah K. To, Frithjof Kruggel, William C. Tang, Andrew W. Browne.

**Funding acquisition:** Andrew W. Browne.

**Investigation:** Jack O. Thomas, Josiah K. To, Frithjof Kruggel, William C. Tang, Andrew W. Browne.

**Methodology:** Jack O. Thomas, Josiah K. To, Parsa Riazi Esfahani, Frithjof Kruggel, William C. Tang, Andrew W. Browne.

**Project administration:** Jack O. Thomas, Josiah K. To, Frithjof Kruggel, William C. Tang, Andrew W. Browne.

**Resources:** Andrew W. Browne.

**Software:** Andrew W. Browne.

**Supervision:** Andrew W. Browne.

**Validation:** Jack O. Thomas, Josiah K. To, Parsa Riazi Esfahani, Frithjof Kruggel, William C. Tang, Andrew W. Browne.

**Visualization:** Jack O. Thomas, Josiah K. To, Parsa Riazi Esfahani, Frithjof Kruggel, William C. Tang, Andrew W. Browne.

**Writing – original draft:** Jack O. Thomas, Josiah K. To, Parsa Riazi Esfahani, Frithjof Kruggel, William C. Tang, Andrew W. Browne.

**Writing – review & editing:** Jack O. Thomas, Josiah K. To, Parsa Riazi Esfahani, Frithjof Kruggel, William C. Tang, Andrew W. Browne.

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
