## [Decision Letter · Decision Letter 0]

22 Oct 2024

PONE-D-24-143843D ophthalmic ultrasonography at the slit lamp using existing ultrasound systemsPLOS ONE

Dear Dr. Thomas,

Thank you for submitting your manuscript to PLOS ONE. After careful consideration, we feel that it has merit but does not fully meet PLOS ONE’s publication criteria as it currently stands. Therefore, we invite you to submit a revised version of the manuscript that addresses the points raised during the review process.

**ACADEMIC EDITOR: **While the manuscript has undergone significant improvements, it can benefit from the current review concerns.==============================

We look forward to receiving your revised manuscript.

Kind regards,

Nader Hussien Lotfy Bayoumi, M.D., FRCS (Glasgow)

Academic Editor

PLOS ONE

Reviewers' comments:

Reviewer's Responses to Questions

**Comments to the Author**

1. Is the manuscript technically sound, and do the data support the conclusions?

Reviewer #1: Partly

Reviewer #2: Yes

Reviewer #3: Yes

2. Has the statistical analysis been performed appropriately and rigorously? 

Reviewer #1: N/A

Reviewer #2: N/A

Reviewer #3: Yes

3. Have the authors made all data underlying the findings in their manuscript fully available?

Reviewer #1: Yes

Reviewer #2: Yes

Reviewer #3: Yes

4. Is the manuscript presented in an intelligible fashion and written in standard English?

Reviewer #1: Yes

Reviewer #2: Yes

Reviewer #3: Yes

5. Review Comments to the Author

Reviewer #1: In the manuscript “3D ophthalmic ultrasonography at the slit lamp using existing ultrasound systems” the authors propose an adapted slit lamp for acquiring 3D ultrasonic images of the eye.

The main concern is that, for my understanding, the same propose has been previously published in: https://iovs.arvojournals.org/article.aspx?articleid=2791001 and this new manuscript lacks originality.

Moreover, the authors argue that previous 3D ophthalmic US scans were not popular in clinical practice and propose a 3D ophthalmic system coupled to a slit lamp as a solution. I’m not sure about the advantages of having the US device coupled to a slit lamp since the two imaging approaches can’t be implemented at the same time.

Related to: “Institutional Review Board approval was obtained from the University of California, Irvine, for clinical data collection.” Isn’t a protocol number?

There is missed information about the image acquisition protocol. How is the probe coupled to the cornea? Do the patients need some preparation for the study? Some information is provided just at the end on the Discussion section, but is should be detailed in M&M.

The comparison of the results obtained with different probes is not relevant (as Fig 4, Fig 5 and related text) since it is intrinsic to the characteristics of the commercial probes. Also, information about the probes is missed, like BW, f-number, focal distance, resolution, etc.

The legend of Fig 7 is incomplete in the draft.

Reviewer #2: The work is genuine, however its application is very difficult and full of challenges.

This paper can be considered a platform to invent a new device, that needs lot of technology and engineering.

The photos are nice but not adding superior information to the usual US or OCT

Reviewer #3: Areas for Improvement:

Lack of Quantitative Analysis:

While the paper provides a qualitative evaluation of the 3D reconstructions, it lacks quantitative measurements or objective performance metrics for assessing the accuracy and resolution of the images produced by the different probes.

Including a detailed comparison of the image resolution, noise levels, and accuracy of the reconstructed 3D models would enhance the scientific rigor of the study and make the findings more robust.

Limited Discussion on Limitations:

The discussion briefly touches on some limitations, such as patient discomfort and motion artifacts during scanning, but these points could be elaborated further. For instance, more attention could be given to the impact of motion artifacts and how they might be mitigated.

Additionally, the limitations of using different ultrasound technologies (CMUT vs. piezoelectric transducers) and their effects on image quality could be discussed in more detail.

Patient Sample Size:

The paper includes in vivo imaging of a small number of subjects, which limits the generalizability of the findings. Future studies should aim to evaluate the system's performance on a larger cohort of patients with a wider variety of ocular conditions to better understand its clinical utility.

Future Research Directions:

While the paper suggests further research is needed, it would benefit from a more detailed outline of the specific next steps. For example, what improvements in the hardware or software could enhance the system? How might machine learning techniques be integrated to automate the interpretation of the 3D images?

6. PLOS authors have the option to publish the peer review history of their article (what does this mean?). If published, this will include your full peer review and any attached files.

Reviewer #1: No

Reviewer #2: **Yes: **Zeinab Elsanabary

Reviewer #3: No

---

## [Author Response · Author response to Decision Letter 0]

18 Nov 2024

We have provided a detailed response to each reviewer’s comments in the attached document titled "Response to Reviewers."

---

## [Decision Letter · Decision Letter 1]

9 Dec 2024

PONE-D-24-14384R13D ophthalmic ultrasonography at the slit lamp using existing ultrasound systemsPLOS ONE

Dear Dr. Thomas,

Thank you for submitting your manuscript to PLOS ONE. After careful consideration, we feel that it has merit but does not fully meet PLOS ONE’s publication criteria as it currently stands. Therefore, we invite you to submit a revised version of the manuscript that addresses the points raised during the review process.

**ACADEMIC EDITOR:**

Thank you for the elaborate clarifications of all the reviewers' concerns. This reviewer though has a few final comments. Please revise.

We look forward to receiving your revised manuscript.

Kind regards,

Nader Hussien Lotfy Bayoumi, M.D., FRCS (Glasgow)

Academic Editor

PLOS ONE

Journal Requirements:

Reviewers' comments:

Reviewer's Responses to Questions

**Comments to the Author**

1. If the authors have adequately addressed your comments raised in a previous round of review and you feel that this manuscript is now acceptable for publication, you may indicate that here to bypass the “Comments to the Author” section, enter your conflict of interest statement in the “Confidential to Editor” section, and submit your "Accept" recommendation.

Reviewer #1: All comments have been addressed

Reviewer #3: All comments have been addressed

2. Is the manuscript technically sound, and do the data support the conclusions?

Reviewer #1: Yes

Reviewer #3: Yes

3. Has the statistical analysis been performed appropriately and rigorously? 

Reviewer #1: N/A

Reviewer #3: Yes

4. Have the authors made all data underlying the findings in their manuscript fully available?

Reviewer #1: (No Response)

Reviewer #3: Yes

5. Is the manuscript presented in an intelligible fashion and written in standard English?

Reviewer #1: Yes

Reviewer #3: Yes

6. Review Comments to the Author

Reviewer #1: All questions were clarified by the author, including the relevance of the method they are presenting.

Reviewer #3: The manuscript is a valuable contribution to the field and is suitable for publication once the following recommended modifications are implemented:

Quantitative Analysis:

Include at least basic quantitative performance metrics, such as resolution or noise levels, to strengthen the scientific rigor.

Probe Specifications:

Add detailed technical specifications of the probes in the Methods section for completeness and transparency.

Expanded Discussion:

Further elaborate on how motion artifacts and patient comfort issues might be mitigated in future iterations of the system.

7. PLOS authors have the option to publish the peer review history of their article (what does this mean?). If published, this will include your full peer review and any attached files.

Reviewer #1: No

Reviewer #3: **Yes: **Ahmed Seddik

---

## [Author Response · Author response to Decision Letter 1]

30 Dec 2024

Please find our detailed response to the reviewers in the attached "Response to Reviewers" file.

---

## [Editor Report · Decision Letter 2]

8 Jan 2025

3D ophthalmic ultrasonography at the slit lamp using existing ultrasound systems

PONE-D-24-14384R2

Dear Dr. Thomas,

We’re pleased to inform you that your manuscript has been judged scientifically suitable for publication and will be formally accepted for publication once it meets all outstanding technical requirements.

Kind regards,

Nader Hussien Lotfy Bayoumi, M.D., FRCS (Glasgow)

Academic Editor

PLOS ONE

Additional Editor Comments (optional):

Thank you for the elaborate responses to reviewers' comments
---

## [Editor Report · Acceptance letter]

19 Jan 2025

PONE-D-24-14384R2 

PLOS ONE

Dear Dr. Browne, 

I'm pleased to inform you that your manuscript has been deemed suitable for publication in PLOS ONE. Congratulations! Your manuscript is now being handed over to our production team.

Kind regards, 

on behalf of

Professor Nader Hussien Lotfy Bayoumi 

Academic Editor

PLOS ONE